# Opportunistic or Non-Random Wildlife Crime? Attractiveness Rather Than Abundance in the Wild Leads to Selective Parrot Poaching

**Pedro Romero-Vidal [1,\*], Fernando Hiraldo [2], Federica Rosseto [3], Guillermo Blanco [4] , Martina Carrete [1] and José L. Tella [2]**

1   Department of Physical, Chemical and Natural Processes, Universidad Pablo de Olavide, 41013 Sevilla, Spain; mcarrete@upo.es
2   Department of Conservation Biology, Doñana Biological Station CSIC, 41092 Sevilla, Spain; hiraldo@ebd.csic.es (F.H.); tella@ebd.csic.es (J.L.T.)
3   Unidad Mixta de Investigacion en Biodiversidad, Universidad de Oviedo, CSIC, 33600 Mieres, Spain; federica.rossetto1991@gmail.com
4   Department of Evolutionary Ecology, Museo Nacional de Ciencias Naturales, CSIC, 28006 Madrid, Spain; gblanco@mncn.csic.es
\*   Correspondence: pedroromerovidal123@gmail.com

**Abstract:** Illegal wildlife trade, which mostly focuses on high-demand species, constitutes a major threat to biodiversity. However, whether poaching is an opportunistic crime within high-demand taxa such as parrots (i.e., harvesting proportional to species availability in the wild), or is selectively focused on particular, more desirable species, is still under debate. Answering this question has important conservation implications because selective poaching can lead to the extinction of some species through overharvesting. However, the challenges of estimating species abundances in the wild have hampered studies on this subject. We conducted a large-scale survey in Colombia to simultaneously estimate the relative abundance of wild parrots through roadside surveys (recording 10,811 individuals from 25 species across 2221 km surveyed) and as household, illegally trapped pets in 282 sampled villages (1179 individuals from 21 species). We used for the first time a selectivity index to test selection on poaching. Results demonstrated that poaching is not opportunistic, but positively selects species based on their attractiveness, defined as a function of species size, coloration, and ability to talk, which is also reflected in their local prices. Our methodological approach, which shows how selection increases the conservation impacts of poaching for parrots, can be applied to other taxa also impacted by harvesting for trade or other purposes.

**Keywords:** CRAVED; conservation criminology; defaunation; harvesting; wildlife trade; parrot abundance; pets; poaching; Savage selectivity index

## 1. Introduction

Defaunation (defined as the global, local or functional extinction of animal populations or species from ecological communities) differs from extinction, as it includes both the disappearance of species as well as their declines in abundance, and has profound ecological consequences, ranging from local to global coextinctions of interacting species to the loss of ecological services critical for humanity [1]. Understanding the causes of defaunation is a growing priority for ecologists, wildlife managers, and conservation biologists, and is important to try to reduce its pace. The drivers of defaunation range from threats operating at global scales, such as climate change, to those that are mostly local, including direct harvest and habitat loss. However, after analyzing the information gathered by the IUCN for

more than 8000 threatened or near-threatened species, Maxwell et al. [2] concluded that by far the biggest drivers of biodiversity decline are overexploitation (the harvesting of species from the wild at rates that cannot be compensated for by reproduction or regrowth) and landscape conversion for food production. Moreover, wildlife overexploitation to meet local and global markets was ranked second of five key drivers of harmful ecosystem change by the Intergovernmental Science-Policy Platform on Biodiversity and Ecosystem Services [3].

Wildlife trade is one of the main causes of overexploitation in some taxonomic groups [1], given that some animal products (e.g., ivory and tiger bones) or groups of species (e.g., cage birds) are highly demanded across the world. Considering closely related species, consumers prefer some over others. For instance, buyers prefer multiflowered species among traded orchids [4], while sale prices of traded songbirds are determined by their body size, coloration and song attractiveness [5]. Body size, coloration and the ability to imitate human speech are traits that make parrots highly valued pets [6], thus making them the most traded vertebrate taxa worldwide [7]. The extent to which these consumer preferences determine poaching activities and their impacts are, however, poorly known. Poachers may supply species according to their availability in the wild or could selectively focus on the most demanded among closely related species. This is a key question with important conservation implications, as selective poaching could cause overexploitation and accelerate the defaunation and even extinction of the most demanded species.

The above question was assessed by wildlife criminologists using the CRAVED model. This model proposes that "hot products" sought by thieves are concealable, removable, available, valuable, enjoyable and disposable (CRAVED) [8] and was applied to data available on the parrot trade in Mexico [9], Bolivia and Peru [10]. These authors concluded that parrot poaching was an opportunistic crime where more widely available species were poached in greater numbers than rare and threatened ones, thus lowering concerns for the conservation impacts of poaching on threatened species [8,10]. However, rough proxies of parrot abundances in the wild were used in these studies, including the number of years each species was allowed to be legally trapped [8] or the detectability of species indicated in field guides [10]. Moreover, these authors recognized that a multivariate approach could have let to different conclusions [8]. Indeed, new analyses [11] challenged these conclusions when applying multivariate analyses to the same data [9], showing that amazons and macaws, the most attractive species as reflected by their body size, coloration and ability to imitate human speech, were disproportionally more traded considering the number of years they were legally trapped, thus contributing to their population declines. Nevertheless, these results could be flawed due to the use of the only available proxies for estimating both wild parrot availability and poaching pressure. The number of years each species was allowed to be legally trapped should reflect their abundance in the wild, assuming that scarcer species were allowed to be trapped for fewer years [8], although international markets, local economics, and political pressures could influence this. On the other hand, the use of seizures as a proxy of poaching pressure [11] may have affected the results, given that seizures usually represent <10% of poaching volumes and are often biased towards certain species [12]. In fact, the proportion of amazons and macaws among all parrots seized in Costa Rica (50%) is significantly higher than among those actually poached and kept as household pets (33%), showing that seizures are biased to the most valuable and threatened species (authors' data, in prep.). Thus, a positive selection of amazons and macaws [11] could at least partially result from seizure biases. Given the limitations of these proxies [8,10,11], reliable information on both the abundance of the species in the wild and poaching pressure is needed to properly test whether parrot poaching is selective or opportunistic [11].

To disentangle whether parrot poaching is a selective or opportunistic activity, we designed a large-scale survey in Colombia, where trapping and keeping native animals as pets is a rooted tradition punished by law since 1977 [13]. We simultaneously measured the relative abundance of 28 parrot species in the wild and as household poached pets. We then applied a selectivity index widely used in habitat selection studies, the Savage index, to ascertain whether pet abundances mirrored wild abundances or, conversely, whether some species were found as pets more than expected. Our results

show a strong selection for more attractive parrot species to be kept as pets, despite their lower abundances in the wild. Positively selected species, but not those less abundant in the wild, were thus the most expensive. This selection has important consequences for the conservation of parrots and their ecosystem services, as well as for our understanding of overharvesting and defaunation, and the management of illegal trade in general.

## 2. Materials and Methods

### 2.1. Study Area

Colombia, with a surface area of 1.1 million km$^2$ and 45.4 million people [14], is one of the most biodiverse countries on Earth [15]. Differences in elevation and latitude produce large climatic variations across the country, which are responsible for the high diversity of habitats. Colombia can be divided into five continental regions (Andean, Caribbean, Pacific, Orinoco, and Amazon), with remarkable biogeographic, socio-cultural, economic, and demographic differences [16]. Using satellite maps, we designed an a priori road itinerary (4232 km in total) to cover the main biomes of the Andean, Caribbean, and Pacific regions across a wide altitudinal range (4–3520 m.a.s.l., Figure 1).

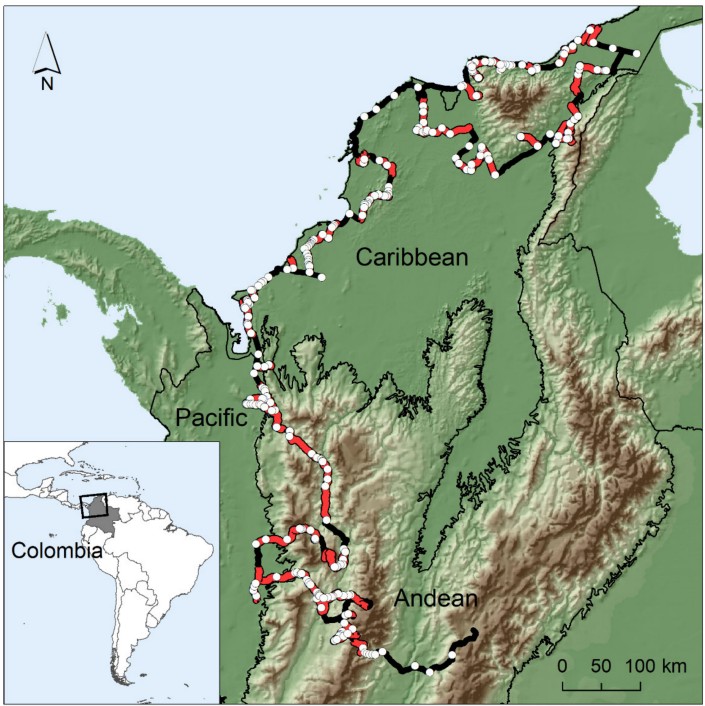

**Figure 1.** Study area showing the itinerary (black line) crossing the Andean, Pacific and Caribbean regions of Colombia, the roadside parrot surveys (in red), and the localities where poached pets were recorded (white dots).

The itinerary (Figure 1) was designed to cover the spatial distribution of many parrot species (35 species in total, see [17]) and to visit villages where we looked for poached pets (see below), thus maximizing the chances of finding a large variety of poached and wild parrot species at a large geographic scale.

### 2.2. Wild Parrot Surveys

We estimated the abundance of parrots in the wild through roadside car surveys, a method adequate for parrots as it allows coverage of large areas, thus increasing the probability of detecting individuals of species occurring at very low densities or spatially aggregated [18–24]. Within the designed itinerary, we selected 2221 km of low-transit and unpaved roads to record all parrots detected

(Figure 1). The beginning and the end of each habitat patch (categorized as pristine natural, degraded natural, mixed, agricultural and urban; see below) was GPS recorded to the nearest 10 m, and the length of the transect varied between 0.1 and 28.88 km (mean = 2.88, SD = 3.63, N = 754). Roadside surveys covered the three different regions surveyed (total rounded lengths: Andean: 839 km, Caribbean: 1002.4 km, Pacific: 379.6 km) and habitats with different degrees of human-induced transformations (see [18]): pristine natural habitats: 31.4 km; degraded natural habitats, where subtle transformations such as selective logging were perceived: 577.5 km; natural habitats mixed with agriculture: 1085.6 km; agriculture: 114.2 km; and rural areas with human settlements: 412.4 km. Surveys were performed only once at each road transect to avoid pseudo-replication and double counting.

　　Surveys were conducted in 2019, during a relatively short period (29 April to 22 May) at the beginning of the wet season, thus avoiding potential spatial biases in parrot abundances due to large seasonal changes. This period mostly coincided with the end of the breeding season, as indicated by the presence of juveniles of several parrot species in the wild, by the full-grown stage of the poached chicks, and by comments of local poachers and pet owners. According to local people, the parrot breeding season was earlier due to weather in 2019.

　　Similar to other roadside parrot surveys [18–24], the driver and two experienced observers drove a 4 × 4 vehicle at low speed (10–40 km/h) from dawn to dusk (aprox. 6 AM–6 PM), avoiding rain and hot middays (from 10:00 to 14:00), when parrot activity decays [25,26], briefly stopping when needed to identify species and to count the number of individuals in flocks. Observers were familiar with parrot species after previous fieldwork in Colombia and surrounding countries, so they were able to visually and aurally identify them. Distances of detection (i.e., the perpendicular distance from parrots to the road when they were detected) were recorded to compare two estimates of parrot abundance (see below). Detection distance was estimated visually for short distances or using a laser rangefinder incorporated into binoculars for large distances (Leica Geovid 10 × 42 HD-R, range: 8–1500 m). In the case of loose flocks, we measured the distance to the closest individual. In some instances, when individuals were heard but not seen and thus flock size could not be estimated, we used the median flock size recorded for the species for analyses. This allowed us to include non-visual contacts, especially of rare and more secretive species, whose omission could result in an underestimation of their relative abundances. All roadside surveys and parrots were recorded using the ObsMapp (Observation International, Amsterdam, The Netherlands) application for smartphones, which uploads the observations to the citizen science platform Observation (www.observation.org, Observation International, Amsterdam, The Netherlands). Therefore, all records, exact location, and associated information can be viewed and downloaded (looking for the observer Pedro Romero Vidal, dates: 29 April–22 May 2019) to be used by other researchers in the future [27].

　　Several methodologies are available to estimate wild parrot abundances, all of them carrying different assumptions, pros and cons, and the adequate method will depend on the objectives of the study [20]. For this work, we used the relative abundance of each species, measured as the total number of individuals recorded divided by the total of km surveyed (indiv./km, [20]). This estimate of abundance has been used in other species- and community-based parrot studies [18–24], with the constraint that it does not account for differences in detectability among species as it is done by distance-sampling method s [22]. However, density estimates through distance sampling are not suitable for our study because (1) they cannot be obtained for rare species with a low number of contacts [22], and thus we could only model densities for 11 of the 28 study species assuming a minimum of 10 contacts per species is enough (but see [20] for the recommendation of using larger numbers of contacts for robust modelling); (2) detection distances cannot be obtained for flocks detected through vocalizations but not sighted, and thus their exclusion would underestimate densities, particularly of the smaller species; and (3) the densities obtained (indiv./km$^2$) cannot be used as units of resource available when applying a poaching selectivity index (see below), while using the total number of individuals recorded allows it. Nonetheless, previous studies on two different parrot communities [19,21] showed that relative abundances (indiv./km) were strongly correlated to detectability-corrected density estimates

(indiv./km$^2$). We measured detection distances for assessing whether it is also the case in this study. We calculated detectability-corrected estimates of parrot densities using the software Distance [28] for 11 species for which densities could be modeled. We assumed that detection decreases monotonically with distance from the road transect [20] and modeled this process using the half-normal detection function [19]. As densities cannot be calculated using non-visual records, we recalculated the relative abundances of these 11 species excluding non-visual records to make results comparable (Table 1). Relative abundances resulted strongly correlated to density estimates (estimate: 0.80, SE = 0.15, t = 5.47, $p$ = 0.0004, adjusted R$^2$: 0.74; Figure 2). These results add further support to the idea that relative abundances of parrots obtained through roadside transects are good proxies of their actual abundance, especially when the high variability in abundance among species overcomes sources of sampling error such as differences in detectability [20].

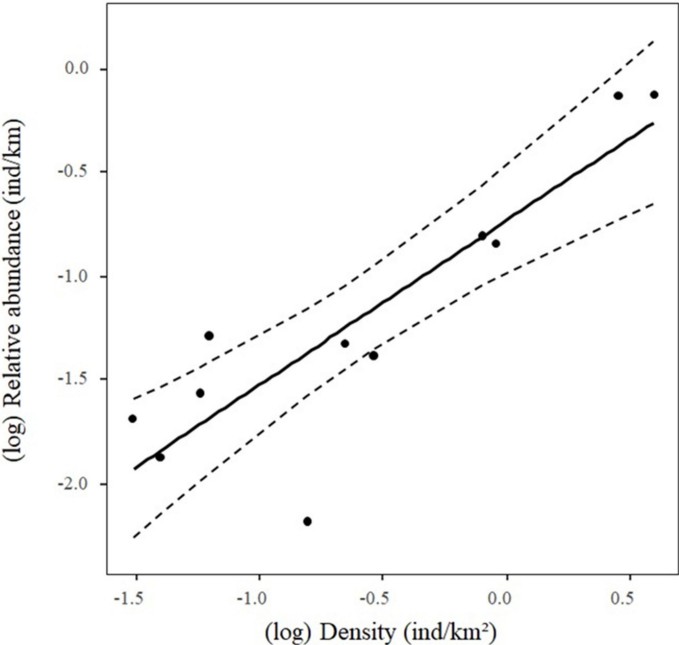

**Figure 2.** Relationship between detectability-corrected estimates of density and relative abundances of 11 parrot species. The regression line (solid) and 95% confidence interval (dashed lines) are plotted.

**Table 1.** Detectability-corrected estimates of density (individuals/km$^2$) obtained through distance sampling and relative abundances (individuals/km) for parrot species with at least 10 visual detections (i.e., sightings of individuals or groups of individuals). Total count refers to the total number of individuals observed during surveys. $w$ indicates the maximum perpendicular detection distance from the survey line for each species.

| Species | Detections | Total Count | Density (ind/km$^2$) | $w$ (m) | Relative Abundance (ind/km) |
|---|---|---|---|---|---|
| *Amazona amazonica* | 24 | 93 | 0.2881 | 200 | 0.0422 |
| *Amazona autumnalis* | 10 | 46 | 0.0307 | 307 | 0.0209 |
| *Amazona ochrocephala* | 43 | 106 | 0.2214 | 348 | 0.0481 |
| *Ara ararauna* | 16 | 60 | 0.0570 | 757 | 0.0272 |
| *Ara severus* | 15 | 30 | 0.0395 | 365 | 0.0136 |
| *Brotogeris jugularis* | 260 | 1678 | 3.9032 | 420 | 0.7620 |
| *Eupsittula pertinax* | 168 | 1669 | 2.7928 | 235 | 0.7579 |
| *Forpus passerinus* | 10 | 15 | 0.1546 | 45 | 0.0068 |
| *Pionus chalcopterus* | 14 | 115 | 0.0621 | 650 | 0.0522 |
| *Pionus menstruus* | 76 | 356 | 0.8029 | 397 | 0.1617 |
| *Psittacara wagleri* | 19 | 322 | 0.9090 | 270 | 0.1462 |

### 2.3. Poached Parrot Surveys

As a direct measure of domestic poaching pressure, we recorded the number of all wild and exotic pets, and how many of them were poached native parrots, in 282 villages crossed by our itinerary (Figure 1). We did not conduct systematic surveys using questionnaires [29], as answers to questions related to illegal activities that are prosecuted in the country would be unreliable [23]. Nonetheless, most people did not hide their pets, nor were they afraid to keep them illegally. Therefore, we recorded many visible pets while driving and walking through streets or entering public establishments, such as shops, hotels or gas stations. We combined these direct observations with informal conversations [23] with randomly chosen local people (N = 358), indicating our interest to see and take pictures of their pets. In about half of the cases (55.6%), people told us they had pets at home, and in 62% of cases provided us with information about other people who poached or owned pets. We then confirmed this information by visiting their homes, taking pictures of the pets (Figure 3) and, at the same time, engaging in informal conversation to obtain additional information, such as the price they paid for the parrot and its ability to imitate human speech (see below). We could not obtain prices from all species, as in many cases the owners did not buy the pets but poached themselves. Pet owners confirmed that all native parrots were poached, with no evidence of attempts to breed them in captivity (contrary to a few exotic, small parrot species). All informants and pet owners shared the information with us freely and were kept anonymous.

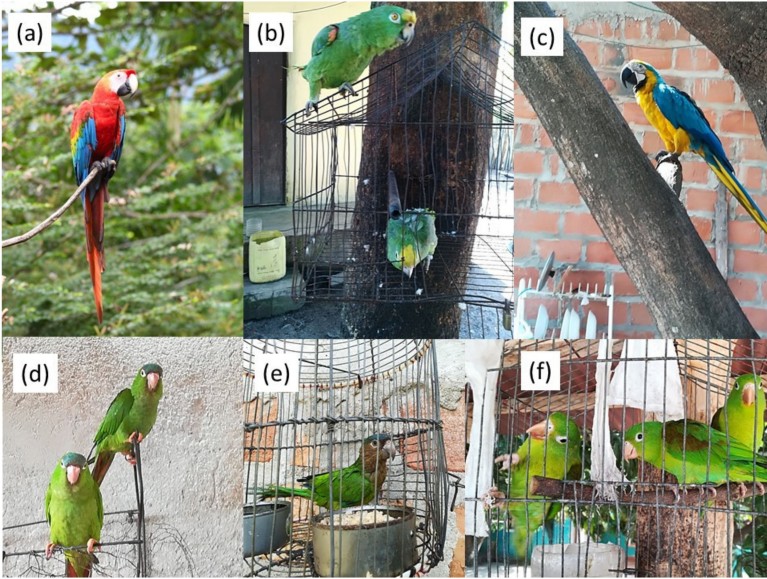

**Figure 3.** Some pet parrots observed in public spaces such as streets (**a**–**c**) or within homes (**d**–**f**). Pictures also illustrate species positively ((**a**): scarlet macaw *Ara macao*, (**b**) yellow-crowned amazon *Amazona ochrocephala*, (**c**) blue-and-yellow macaw *Ara ararauna*, (**d**) blue-crowned parakeet *Thectocercus acuticaudatus*) and negatively selected by people as pets ((**e**) brown-throated parakeet *Eupsittula pertinax*, and (**f**) orange-chinned parakeet *Brotogeris jugularis*). Pictures: P. Romero-Vidal and J.L. Tella.

### 2.4. Parrot Attractiveness

The attractiveness of each parrot species was rated based on its body size (obtained from [30]), coloration, and ability to imitate human speech [8,11]. Parrot coloration was described as the proportion of the body (bright body) and head (bright head) covered by bright colors (i.e., other than the dominant green or brown coloration), scored from 0 to 2 following [8], and the total number of colors (N colors) observed when the bird is perched, using plates in [29]. The ability of each individual pet to imitate human speech was ranked into five categories using the information provided by local pet owners (0: individuals not able to make imitations, 0.5: individuals able to whistle or imitate one or two words, 1: individuals able to imitate several words but poorly pronounced, 1.5: individuals able to imitate

several words, with good pronunciation, and 2: individuals able to imitate human speech, using a wide repertoire of words and making up short sentences, singing songs, imitating other domestic animals or sounds such as telephone, TV, radio, etc.). Scores from different individual pets were averaged within species to obtain a rank describing the ability of each species to speak. However, the opinion of local pet owners could be biased by their experience, which is usually limited to their own pets. Therefore, we asked the same question to five people from USA, France, Germany and Spain with >20 years of experience breeding and keeping a large variety of parrot species in captivity. The average scores provided by these experts correlated well with those provided by local pet owners (Spearman rank correlation = 0.85, *p* < 0.0001, Figure 4), thus validating the use of local knowledge for measuring the mimicry ability of different parrot species.

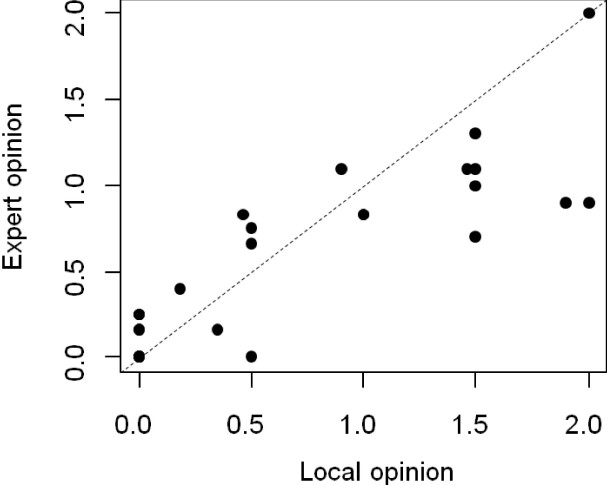

**Figure 4.** Scatterplot of the averaged opinion on the ability of different parrot species from Colombia to imitate human speech (0 = lowest, 2 = highest) provided by international experts and by local pet owners. Each dot represents a species, and the dashed line represents the theoretic perfect correlation.

*2.5. Statistical Analyses*

We used the Savage selectivity index [31] to assess whether parrot species are poached proportionally to their abundances in the wild. This index is widely used in resource selection studies (e.g., [32]) and allows us to infer the statistical significance of selection [31]. We used the number of parrots of each species recorded in the wild as units of resource availability and numbers recorded as pets as units of the resource used. The Savage selectivity index was calculated for each species as $W = U_i/p_i$, where $U_i$ is the proportion of a given species (among all poached parrots) recorded as a pet (i.e., used) and $p_i$ is the proportion of that species (among all wild parrots) recorded in the wild (i.e., available). A few species were so scarce that we did not record a single individual in the wild (Table 2) despite their known presence in the study area [17] and finding them locally as pets. In those cases, as the availability of a used resource cannot be zero [31], we conservatively considered that at least one wild individual was recorded to allow calculating the Savage index. The Savage index theoretically varies from zero (full negative selection) to infinite (full positive selection), with one being the expected value by chance (i.e., used in proportion to its availability) [31]. The statistical significance of this index is obtained by comparing the statistic $(wi - 1)^2/se_{wi}^2$ with the corresponding critical value of a $\chi^2$ distribution with one degree of freedom [31], the null hypothesis being that species are poached in proportion to their availability in the wild. The standard error of the index ($se_{wi}$) is calculated as $\sqrt{[(1 - p_i)/(u_+ \times p_i)]}$, where $u_+$ is the total number of poached parrots recorded. Statistical significance was obtained after applying the Bonferroni correction for multiple tests. We did not calculate the Savage index for four species (Table 2) since they are unable to survive in captivity more than a few days or weeks [6] and thus they are rarely poached. However, results including these species were nearly identical (Spearman correlation, r = 1, *p* < 0.0001, *n* = 24).

**Table 2.** Parrot species included in the study, their body size (in cm), the scores (0–2) for the brightness of body and head coloration, the total number of colors (Color), their ability to imitate human speech (speech, 0–2, with ranges), their price in US$ (with SD), the number of individuals recorded in the wild (N wild) and as poached pets (N pet), and the Savage selectivity index (W). * Statistically significant W values after Bonferroni correction (*p* < 0.002).

| Species | Size | Body | Head | Color | Speech | Price | N Wild | N Pet | W | |
|---|---|---|---|---|---|---|---|---|---|---|
| *Amazona amazonica* (Aam) | 31 | 0.2 | 1 | 3 | 1.5 (1–2) | 34.85 (-) | 93 | 12 | 1.17 | |
| *Amazona autumnalis* (Aau) | 34 | 0.2 | 1 | 3 | 1.5 (1–2) | | 61 | 25 | 3.73 | * |
| *Amazona farinosa* (Afa) | 38 | 0.2 | 0.2 | 1 | 1.9 (1–2) | 38.72 (13.4) | 20 | 18 | 8.18 | * |
| *Amazona mercenarius* (Ame) | 34 | 0.2 | 0 | 1 | 1.0 (1–1) | | 93 | 0 | 0.00 | * |
| *Amazona ochrocephala* (Aoc) | 31 | 0.2 | 0.8 | 2 | 2.0 (1–2) | 34.41 (16.0) | 136 | 359 | 24.00 | * |
| *Ara ambiguus* (Aab) | 85 | 1.5 | 0.5 | 3 | 1.5 (1–2) | | 0 | 3 | 27.28 | * |
| *Ara ararauna* (Aar) | 85 | 2 | 1.8 | 3 | 0.9 (0–2) | 43.57 (20.5) | 80 | 76 | 8.64 | * |
| *Ara chloropterus* (Ach) | 90 | 2 | 2 | 3 | 1.5 (1–2) | | 0 | 14 | 127.28 | * |
| *Ara macao* (Ama) | 85 | 2 | 2 | 3 | 2.0 (2–2) | 145.22 (0.00) | 4 | 74 | 168.20 | * |
| *Ara militaris* (Ami) | 75 | 1.5 | 0.5 | 3 | 1.5 (0–2) | | 10 | 7 | 6.36 | * |
| *Ara severus* (Ase) | 46 | 1 | 0 | 2 | 1.0 (1–1) | | 54 | 5 | 0.84 | |
| *Bolborhynchus lineola* (Blin) | 16 | 0 | 0 | 1 | 0.0 (0–0) | | 1 | 0 | 0.00 | |
| *Brotogeris jugularis* (Bju) | 18 | 0 | 0 | 1 | 0.4 (0–2) | 6.53 (1.45) | 6230 | 344 | 0.50 | * |
| *Eupsittula pertinax* (Epe) | 25 | 0.1 | 0 | 1 | 0.5 (0–2) | 5.68 (3.67) | 2445 | 189 | 0.70 | * |
| *Forpus conspicillatus* (Fco) | 12 | 0.1 | 0 | 1 | 0.0 (0–0) | | 83 | 0 | 0.00 | |
| *Forpus passerinus* (Fpa) | 12 | 0.1 | 0 | 1 | 0.0 (0–0) | 5.81 (0.00) | 35 | 6 | 1.56 | |
| *Forpus spengeli* (Fsp) | 12 | 0.1 | 0 | 1 | 0.0 (0–0) | | 80 | 9 | 1.02 | |
| *Hapalopsittaca fuertesi* (Hfu) | 23 | 0.5 | 0.9 | 3 | 0.0 (0–0) | | 1 | 0 | - | |
| *Ognorhynchus icterotis* (Oic) | 42 | 0.8 | 0.8 | 2 | 0.5 (0–1) | | 85 | 2 | 0.21 | |
| *Pionus chalcopterus* (Pch) | 29 | 0.2 | 0 | 2 | 0.0 (0–0) | | 134 | 3 | 0.21 | |
| *Pionus menstruus* (Pme) | 28 | 0.3 | 1.5 | 2 | 0.2 (0–2) | 16.46 (9.22) | 497 | 19 | 0.35 | * |
| *Pionus seniloides* (Pse) | 30 | 0.2 | 0 | 2 | 0.0 (0–0) | | 2 | 0 | 0.00 | |
| *Pionus sordidus* (Pso) | 28 | 0.2 | 0 | 1 | 0.0 (0–0) | | 17 | 1 | 0.52 | |
| *Psittacara wagleri* (Pwa) | 36 | 0 | 0.5 | 2 | 0.5 (0–1) | | 628 | 10 | 0.14 | * |
| *Pyrilia haematotis* (Pha) | 21 | 0.1 | 0 | 2 | 0.0 (0–0) | | 0 | 1 | - | |
| *Pyrilia pyrilia* (Ppy) | 24 | 0.1 | 2 | 3 | 0.0 (0–0) | | 1 | 0 | - | |
| *Psittacara wagleri* | | | | | | | | | | |
| *Thectocercus acuticaudatus* (Tac) | 37 | 0 | 0.1 | 1 | 0.5 (0–1) | | 20 | 13 | 5.91 | * |
| *Touit batavicus* (Tba) | 14 | 0.1 | 0 | 3 | 0.0 (0–0) | | 1 | 0 | - | |

We used principal component analysis (PCA) to obtain a composite variable that describes the attractiveness of each parrot species as a function of its color, body size, and ability to speak. Variables, which were positively correlated (Pearson correlations: 0.50–0.94, all *p* < 0.0001), were scaled before analysis. Bartlett's test of sphericity was computed to establish the validity of the data set. Eigenvalues > 1 were used to assess the number of factors to extract.

We used generalized linear models (GLMs) to test whether the preference of people for certain species (measured through the Savage index; log-transformed; normal error distribution and identity link function) was related to their attractiveness. We then assessed whether preferred or rare species were the most valuable in monetary terms by relating the Savage indexes and abundances of species in the wild (ind./km) to their price (log-transformed; normal error distribution and identity link function). We used the average prices of species provided by pet owners (local currency transformed to US$, Table 2). All statistical analyses were performed in the R v.3.6.1 statistical platform [33].

## 3. Results

We recorded 10,811 wild individuals from 25 parrot species (Table 1) across the 2221 km of roadside surveys conducted, covering a wide variety of biomes with different degrees of human alteration. Overall abundance reached 4.87 ind./km, although most records (80.31%) corresponded to just two parakeets (orange-chinned parakeet *Brotogeris jugularis* and brown-throated parakeet *Eupsittula pertinax*). The other species were present in low numbers, were extremely rare or even unrecorded in the wild (Table 2). Simultaneously, we recorded a total of 2465 pets from 124 species, kept by 818 owners in 92.9% of the 282 villages surveyed (Figure 1), from which 1179 (47.8%) were pets from 21 native parrot species (Table 2). The rest of the pets were mostly songbirds (Passeriformes, 32.7%), non-native parrots (12.7%), other birds (3.0%), mammals (1.3%), and reptiles (0.4%). Among the 358 local people we met, 58.4% of them kept poached native parrot pets at the time of our survey or recently, and 38.0% knew other people also keeping them.

In absolute numbers, *B. jugularis* and *E. pertinax* made up almost half (45.20%) of all pet parrots. However, these species were actually negatively selected when considering their abundances in the wild (Table 2, Figure 5). On the contrary, most amazons (*Amazona* spp.), large macaws (*Ara* spp.) and *Thectocercus acuticaudatus*, mostly uncommon or extremely rare in the wild, were strongly positively selected as pets (significant W > 1). The other species showed non-significant selection (i.e., were kept as pets in proportion to their availability in the wild; Table 2, Figure 5).

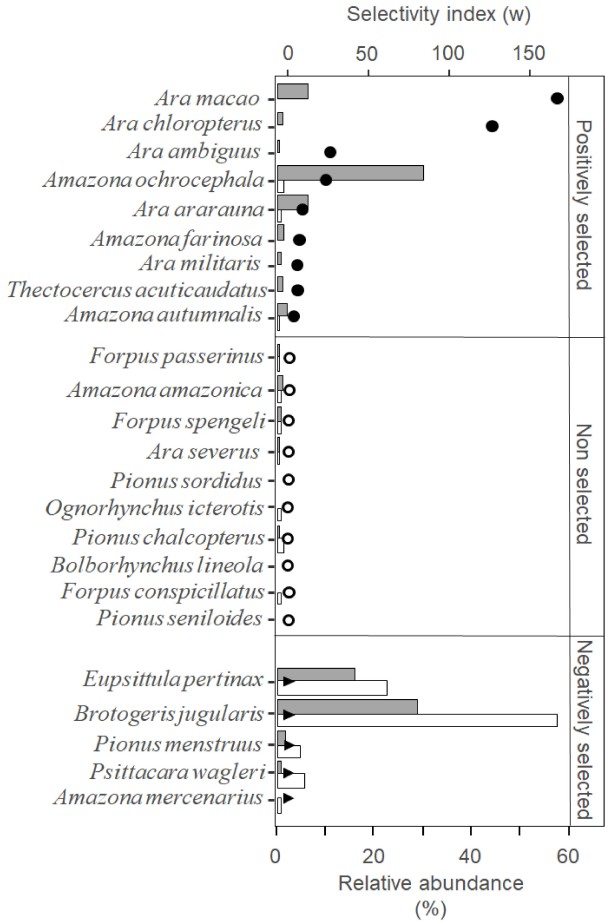

**Figure 5.** Relative abundance of parrots in Colombia as pets (dark gray bars) and in the wild (white bars), and Savage selectivity index (W; black dots: significant positive selection; black triangles: significant negative selection; white dots: not selected).

The PCA analyses (Bartlett's Test of Sphericity: $\chi^2 = 99.24$, $p < 0.0001$, df = 10) rendered a single dimension with an eigenvalue > 1 (3.40), which positively correlated with body size (0.92), coloration (bright body: 0.91, bright head: 0.79, number of colors: 0.77) and ability to imitate human speech (0.71), explaining 68.08% of the total variance. Thus, PC1 can be interpreted as a descriptor of parrot attractiveness, large, colorful and talkative species being more attractive (positive values) than their counterparts (negative values; Figure 6).

PC1 was positively related to the Savage index (estimate: 0.27, SE: 0.04, t = 6.40, $p < 0.0001$, adjusted-$R^2 = 0.63$), showing that the most attractive species were poached in larger numbers than expected based on their availability in the wild (Figure 7a). The price of the species increased with their attractiveness (estimate: 16.24, SE: 4.06, t = 4.00, $p < 0.0052$, adjusted-$R^2 = 0.65$, Figure 7b) but was unrelated to their abundances in the wild (estimate: 0.85, SE: 0.78, t = −1.09, $p = 0.3139$, Figure 7c), indicating that the most attractive but not the rarest species were more valuable.

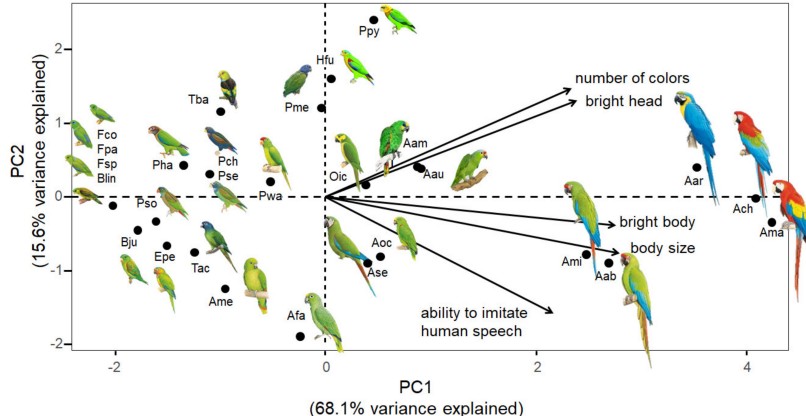

**Figure 6.** Principal component analysis (PCA) of Colombian parrot traits, namely: body size, coloration (bright body, bright head and number of colors) and ability to imitate human speech. See Tabl 2 for species abbreviations. Drawings of parrots are not scaled. PC2 is plotted to allow better visualization of species across the PC1 axis, which reflects parrot attractiveness, but was not used for further analysis (eigenvalue < 1).

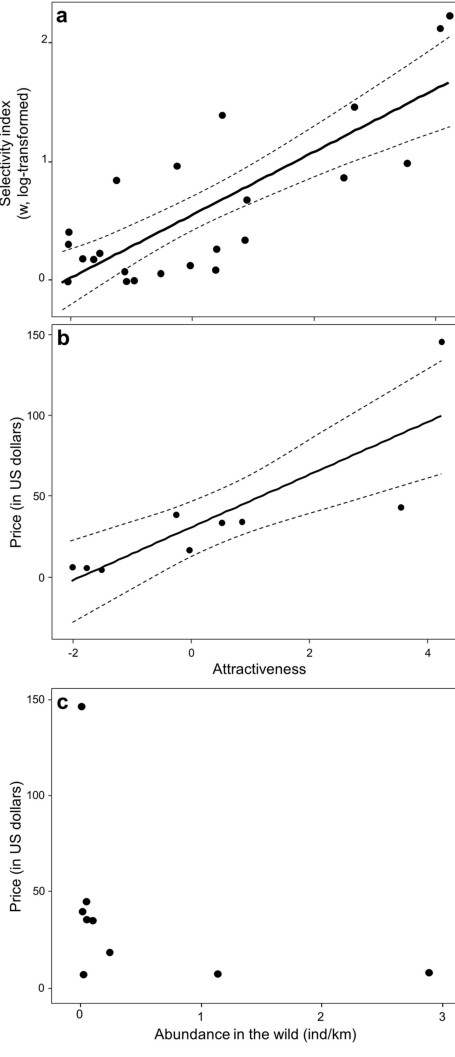

**Figure 7.** Preferred parrots in Colombia (measured through their Savage selective index) were the most attractive (i.e., large, colorful, and able to imitate human speech) species (**a**), which were also the most expensive (**b**), independently of their abundance in the wild (**c**).

## 4. Discussion

### 4.1. Parrot Poaching Is Not an Opportunistic, but a Selective Wildlife Crime

Wildlife trafficking is increasingly recognized as both a specialized area of organized crime and a significant threat to many plant and animal species [2,34,35]. However, due to its intrinsic illegal nature, it is difficult to fully know its actual extent and consequences for wildlife [12,36]. Here, we provide the first reliable and simultaneous large-scale estimation of poaching pressure and abundance in the wild of a community of parrot species, showing that poaching of this taxonomic group is not opportunistic, but largely focused on species with particular traits that make them more attractive to people. Following the CRAVED model approach [8], our data would have suggested that parrot poaching is an opportunistic crime, as the numbers of poached parrots per species positively correlates with numbers recorded in the wild (Kendall's Tau-b = 0.33, $p$ = 0.018, N = 27). However, this slight trend, which is markedly influenced by two parakeet species (which jointly made up >80% and >45% of the individuals recorded in the wild and as poached pets, respectively), turns out to be non-significant when they are removed from the analysis (Kendall's Tau-b = 0.23, $p$ = 0.125, N = 25). Moreover, as recognized by authors using the CRAVED model [8], conclusions derived from simple univariate analyses could change when simultaneously testing the effects of other variables, such as species attractiveness in multivariate models. This possibility was later confirmed when reanalyzing the same parrot poaching data from Mexico using generalized linear models: attractive species were more poached than expected when controlling for the number of years they were allowed to be legally trapped [11].

To identify selection, it is not only important to assess resource availability, but also be able to calculate its statistical significance. Here, we provide direct estimates of parrot availability and poaching and a key analytical advance, the application of the Savage selectivity index [31] to quantitatively measure poaching selection. To our knowledge, this is the first time that a selectivity index is used to statistically evaluate whether any given species is positively, negatively, or not selected at all. A further advantage of this index is that it can be used as a continuous response variable to ascertain drivers of poaching selection. In this sense, we found that 63% of the variance in parrot poaching selection is explained by the attractiveness of the species, thus confirming that poaching is not a taxonomically random, but a species-specific activity that preferentially focuses on the most attractive species for people [11].

Attractiveness in this taxonomic group has been found related to body size and coloration, which determines for instance which species are kept in zoos [37]. Meanwhile, the ability to imitate human speech can be particularly appealing when parrots are kept as pets at close contact with people [6]. Thus, the combination of these traits can describe species attractiveness and, therefore, predict their selection as pets and their prices [11]. As found in other countries [11,38,39], in Colombia macaws and amazons were much more expensive than other poached parrot species. We also show that higher prices in the domestic pet trade are not related to the rarity of a species in the wild, but strongly related (65% of the variance explained) to its attractiveness. While both rarity and physical attractiveness influence the prices of internationally traded birds [40], our results show that local demand focuses on attractive rather than on rare parrot species.

The quantitative measurement of poaching selection also allows deeper investigation of the unexpected preference of some species and additional cultural drivers of selection. For example, the high positive selection of the scarlet macaw *Ara macao* among Colombian people is surprising (Table 2), while its attractiveness is not much higher than that of similar macaw species (Figure 6). Local knowledge provided us with the answer: people explained to us that this species is sought after because its plumage resembles the Colombian national flag, hence its local name "guacamaya bandera" (flag macaw). Moreover, they also described that the Colombian guerrillas (revolutionary armed forces) persecuted its capture and use as pets because it was considered as unpatriotic; thus, poaching pressure on this species has increased since the guerrillas ceased their warlike activities.

As this species became extremely rare because of overharvesting [17], poachers seemed to switch efforts towards the similarly sized and colored green-winged macaw *Ara chloropterus* (Figure 6) as a substitute species, thus also explaining its outstanding selection (Table 2). Another case that merits attention is the positive selection of the blue-crowned parakeet *T. acuticaudatus* (Table 2), despite its low attractiveness rating (Figure 6). This species is restricted to very dry forests of the Guajira region, where the most preferred species such as macaws and amazons are absent [17], and thus it is the largest and most colorful species available. Other potential covariates of poaching selection could be assessed in further studies, such as the accessibility of nests and life expectancy of parrot species as pets.

*4.2. Conservation Implications of Selective Parrot Poaching*

The colorful plumage of parrots and their ability to imitate human speech have made them highly popular as pets [6], thus leading to the international trade of at least 259 species of parrots, involving millions of individuals in recent decades [7,41,42]. In the near absence of long-term monitoring programs of wild populations [43] and analyses of sustainable harvesting [44,45], international trade of wild-caught individuals may constitute a threat to many parrot species worldwide [11,46,47]. A concerning example is the African grey parrot *Psittacus erithacus*, considered the best at imitating human speech among all extant parrot species [6]. Overharvesting due to trapping for the international trade has caused large range contractions and decimated the populations, to the point that the species was included in Appendix 1 of CITES in 2017, prohibiting international trade on wild specimens for commercial purposes, and was listed as globally Endangered by IUCN in 2018 [48]. Although international bans have largely reduced the legal trade on parrots [42,49] and the upsurge of captive-breeding [6,49] has reduced the demand of wild-caught traded birds, illegal trade is still active [50], although at much lower volumes, including illegal trade on African grey parrots [51]. Nonetheless, while international trade is a matter of concern, less attention has been paid to the conservation impact of domestic trade on parrots, even though it is known to occur in different regions of the world, such as Madagascar [52], Asia [34], and all across the Neotropical region [9,23,35,38,39,53–57]. Due to its illegal nature, the true scale and impact of parrot poaching are often underestimated [34] and based mainly on counts from pet markets [53,55,58], government seizures, or other information sources difficult to verify [9,35,53].

In the Neotropics, expert knowledge indicated that 68% of the studied parrot populations are threatened due to their capture for the domestic pet trade [59]. However, it is unknown whether and to what extent poaching threatens these species differentially. Based on conclusions obtained through the CRAVED model, parrot poaching would mostly affect common species the most, thus alleviating concerns on its conservation impacts [8,10,60] since harvesting individuals of common species could be even considered as sustainable resource use [55,61,62]. Our results lead to the opposite conclusion: the two most common species are poached half as much as expected based on their availability in the wild, while a few, highly attractive species are poached in larger numbers than expected according to their abundance and are likely to be overharvested. Pet owners indicated that these still abundant parakeets are poached as substitutes of more preferred species, when the former are not available because of their scarcity in the wild and/or their high prices. Therefore, our concern is not the absolute number of individuals poached of a given species but the proportion of the wild population size. In fact, the poaching of as few as 70 individuals per year constitutes a major threat to the Critically Endangered red siskin *Spinus cucullatus* in Venezuela [63]. The trade of some attractive parrot species has been shown to cause negative population trends and affect their conservation status [11]. A proper test of the overharvesting effects of parrot poaching would be to relate the selection on each species to their population trends. While detailed information on population trends is not available for Colombian parrots, they could be estimated through expert knowledge [58]. However, there is evidence that poaching has caused large population declines and range contractions of the yellow-crowned amazon *Amazona ochrocephala*, considered as the species that best imitates human speech in Colombia [17], and of the highly demanded scarlet macaw, while species we identified as less preferred or not selected

at all by poachers have not suffered large declines in Colombia [17]. It is worth mentioning that the scarlet macaw was considered the most abundant macaw species in the region in the 1950s, in contrast with its current rarity [17] in this study.

Although adult parrots are also eventually trapped [64], parrot poaching mostly focuses on nestlings [38,64,65] as hand-reared chicks make better pets than birds caught as adults in terms of docility and ability to learn human speech [6]. This has different implications on the population dynamics of the poached species [64], as lifespan generally increases with the size of parrot species [66]. Nest poaching of the largest species such as amazons and large macaws could alleviate concerns about its impact as they have the longest lifespans among parrots (at least 34–63 years in captivity [66]), and thus small reductions in their breeding success due to poaching could have less impact on their population dynamics compared to small, short-lived species. However, we learned from local people that the last remaining nests of the preferred amazon and macaw species are located and poached year after year, often for decades. Indeed, local poachers compete for the same nests to the point that nests are surveyed daily to avoid robbing by others. Therefore, breeding pairs may occupy the same areas for decades, giving the wrong impression of apparent population stability to local people acting as regular or occasional poachers, birdwatchers, and wildlife managers. Ultimately, if current poaching pressure is not halted, the remaining populations will collapse due to the senescence and death of breeding adults in the absence of population recruitment.

### 4.3. Ecological Implications of Selective Parrot Poaching

Selective parrot poaching severely affects the conservation of the preferred species, as well as the ecosystem services they provide. Parrots have been long considered as plant antagonists, given their undoubted role as seed predators [67]. However, they also provide several ecological functions [68] within an antagonist-mutualism continuum [69]. Particularly, parrots can act as effective seed dispersers through complementary mechanisms, such as stomatochory [21,23,70–73], endozoochory [24,74,75], and epizoochory [76], further facilitating secondary seed dispersal by a variety of other species [77]. Altogether, they may play an important role in the structure of networks, communities, and ecosystems [19,21,69]. Poaching reduces the population size of parrots, thereby quantitatively reducing and threatening their ecological functions. In fact, the selective poaching of the largest species (amazons and macaws) may have the strongest impact, as these species are the main—and sometimes the only—effective long-distance seed dispersers of palms and trees with large-sized fruits, which are biomass-dominant and key species in several ecosystems [21,71–73]. The defaunation of these large-sized parrot species, which could be considered as megafauna attending to a new functional definition [78], further reduces the dispersal of large-fruited plants that previously was only attributed to the decimated large-sized mammals and those extinct in the Pleistocene in South America [72]. The dispersal of some of these tree and palm species has already been disrupted after the large-scale extirpation and population declines of some amazon and macaw species [20,73].

### 4.4. Suggested Conservation Actions

Keeping parrots as pets in Colombia, as in other Neotropical countries, seems to be ancestrally rooted [17]. This cultural tradition could have been sustainable in the past but not today, given the large human population and economic power increase in recent decades [14]. These two factors have increased the demand for pets while promoting habitat loss, also affecting parrot populations [17]. Therefore, conservation actions are urgently needed to halt parrot defaunation. Based on the conclusions derived from CRAVED model analyses, conservation actions should focus on the most heavily poached species by protecting and preventing poaching in their breeding areas [8,10,60]. In Colombia, this would mostly apply to two parakeet species with a wide distribution [17], thus making the protection of breeding sites unfeasible. Moreover, these species have large, non-threatened populations [17], and thus their conservation should be not a priority. Our results on selective poaching provide a completely different conservation management scenario, as actions must focus on the most preferred, currently

overexploited species. The protection of breeding sites to avoid nest poaching [8,10,60] may be efficient in the case of species with restricted breeding ranges [18,79,80], but it is not feasible for Colombian macaws and amazons, with large distribution ranges and low population densities [17] in this study. The attraction of ecotourism and the creation of eco-lodges may increase local incomes and reduce poaching [60], and favor research and the conservation of large parrots and macaws [81]. Colombia has great potential and should promote these conservation-friendly economic activities, but these local activities cannot prevent parrot poaching at a national scale. Paradoxically, in the absence of law enforcement in Colombia, we found tourist establishments displaying captive macaws to attract tourists.

As in other Neotropical countries [82], we learnt from pet owners that parrot poaching in Colombia is generally not an organized crime, but is performed by local people to obtain their own pets or supply pets to neighbors and relatives. A large proportion of the population (c. 60%) is involved in the illegal activity of keeping native parrots as pets, often acting simultaneously as poachers and consumers, and this activity is widespread across the country. Law enforcement and reducing the demand are two strategies to reduce wildlife poaching that must be balanced in terms of cost-effectiveness, especially when conservation resources are limited [83]. Police control should be strengthened to dissuade people from keeping pets at least of overexploited species, while educational campaigns for public awareness on the consequences of poaching should reduce the demand [18]. Alternative sources can also be offered to satisfy the cultural tradition of owning pet parrots. Breeding parrots in captivity is well established [6], and can successfully supply the previous demand for internationally traded wild parrots [49]. Thus, breeding native parrots for local sale [9] could reduce the pressure on wild populations. However, the low-reproductive rates of preferred species (amazons and macaws) in captivity [6] make it difficult to supply enough individuals and at prices low enough to counteract poaching. Moreover, this activity is prone to fraud, as chicks of preferred species could be poached and sold as captive bred (see [84] for traded Asian songbirds). The genetic control of supposedly captive-bred individuals [85] requires great surveillance efforts, the development of genetic markers, and the availability of molecular laboratories [86], which are difficult to implement at a large scale in countries such as Colombia.

An alternative is to supply the pet demand with captive-bred exotic parrot species that are easier to reproduce [6], can be bought at competitive prices, and show low risks of invasion when they are accidentally released to the wild [87,88]. A combination of these actions seems to have been successful at halting parrot poaching in a small Colombian region, within the Andean distribution of the yellow-eared parrot *Ognorhynchus icterotis*, a globally endangered species for which conservation programs and awareness campaigns were implemented over decades [89]. In this area, most people have non-native pet parrots, such as budgerigars *Melopsittacus undulatus*, cockatiels *Nymphicus hollandicus* and lovebirds *Agapornis* spp., often after the seizure of their native pets by the police, and are very aware of the illegal nature of this activity. Wildlife authorities should realize that law enforcement and demand reduction must be urgently extended to the whole country to avoid, at least, the predicted population collapse of overexploited species. Considering cost-effectiveness [83], law enforcement is probably the most effective action at a national scale, since police are widespread across the country and should simply apply current laws without the need for additional economic costs. However, the seizure of all parrot pets is unfeasible due to the economic costs of creating and maintaining wildlife rescue centers [35,65] to hold them. In fact, seized birds are often returned to the wild to reduce costs and to create space for newly confiscated individuals, in the absence of reintroduction programs [35]. Thus, seizures should focus on overexploited species and should be combined with well-designed awareness campaigns [90,91] to reduce demand. On the other hand, captive breeding of exotic parrots to supply the demand should also be promoted, but under strict sanitary control, as exotic parrots can carry pathogens (e.g., [92]) that could spread and negatively impact native populations.

*4.5. Further Prospects for Assessing Selective Harvesting*

Solving the dichotomy between opportunistic or selective poaching has profound conservation implications, since the overexploitation of preferred species may be causing their decline, pushing their populations toward regional [11,58] and global [84] extinctions. Several lines of evidence show that any form of harvesting (including legal fishing and hunting) is selective toward individuals of a certain sex, size, morphology, or behavior, with long-term population and evolutionary consequences [93–99]. However, to our knowledge, the hypothesis of selective harvest at the community level (i.e., on species with particular characteristics over others) has not yet been properly tested, mainly due to the difficulty of assessing their availability in the wild. The application of a selectivity index allows a quantitative measure and statistical test of harvesting selection in both intra- and interspecific studies. Therefore, it is a powerful tool for assessing selection and investigating the factors driving it, not only in other poached parrot communities and heavily traded birds, such as Asian songbirds [58,84,100,101], but also in other animal and plant species harvested, for example, through deforestation, fisheries, game hunting, or bush-meat exploitation.

**Author Contributions:** Conceptualization, J.L.T., P.R.-V., G.B., F.H. and M.C.; methodology, P.R.-V., J.L.T. and F.H.; statistical analysis, P.R.-V., F.R. and M.C.; writing—original draft preparation, P.R.-V., M.C. and J.L.T.; writing—review and editing, J.L.T., M.C., P.R.-V., G.B. and F.H.; visualization, P.R.-V., M.C., F.R., J.L.T., G.B. and F.H.; supervision, J.L.T. and M.C.; project administration, M.C.; funding acquisition, J.L.T. and M.C. All authors have read and agreed to the published version of the manuscript.

**Funding:** This research was funded by Loro Parque Fundación (PP-146-2018-1).

**Acknowledgments:** E. Arrondo, I. Paredes, L. López-Ricarte and M.C. Díaz helped during fieldwork, and I. Afán (LAST-EBD) elaborated the maps. Logistic and technical support were provided by Doñana ICTS-RBD. Two anonymous reviewers greatly helped to improve the manuscript. This work did not require approval by our Ethical Committee (EBD-CSIC) since it did not involve invasive methods nor experimental work with live animals.

**Conflicts of Interest:** The authors declare no conflict of interest.

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
