# Peer review of "Opportunistic or Non-Random Wildlife Crime? Attractiveness Rather Than Abundance in the Wild Leads to Selective Parrot Poaching"

_diversity, doi:10.3390/d12080314_

Round 1
Reviewer 1 Report
Opportunistic or non-random wildlife crime? Attractiveness rather than abundance in the wild leads to selective parrot poaching
The manuscript deal with poaching on Neotropical parrots in Colombia. Comparing data from road survey (as proxy for parrot’s availability) and birds kept as pets (assuming poached) in roadside villages authors tried to answer the question whether poaching is rather opportunistic or selective. Additionally analyses connected with particular species attractiveness, as well as their prices are presented. Analyses are based on large sample sizes, lots of information is presented among other how local peoples treated poaching and wildlife in general.
Especially discussion is valuable, presenting not only the problem of parrots poaching itself, but presenting poaching and its importance in wider, ecosystems context. Some information presenting in discussion – e.g. suggestions how deal with poaching, replacements of native birds by exotic, captive breeding birds as a methods of decreasing poaching pressure etc. will be useful to present, or at least mentioned in the abstract, as it is valuable for parrots conservation, so important for wider audience.
The paper is well designed, analysed and written, and in my opinion only minor corrections are necessary.
Minor comments:
Authors – at least in discussion - should mention that the way/how easy birds could be caught in the wild should be also considered as a factor affecting poaching and species are kept in villages. Maybe birds that could be caught easier will be kept more frequently? This could be also the reason why large parrots are so popular among poached birds (besides well presented other – probably more important - factors). They need large cavities to breed, probably the same cavity is used for several years (as it is mentioned in discussion) and this also is related with availability of birds for poachers.
Maybe also large birds which live longer – due to longevity - could be more likely to be kept.
Abstract:
- 29 – probably ‘local prices’ will be better.
Introduction
There is lack of information why authors assume that birds kept in villages are from poaching (could be used as a proxy for poachers impact on birds), but not from e.g. breeding of animals kept for long time. This information appears in discussion, but it should be clear, and it will be presented in introduction (or methods).
Lines 95-100 is looks quite strange when main results appears in introduction. In my opinion rather some expectations/hypotheses should be presented here.
.
Methods:
Are authors assume that distribution of parrot species is potentially similar across the country (or at least along roads), or some species occur only in particular areas?
Line 125 – it is mention that surveys were carried out at the beginning of wet season. What is the phase of birds’ life? Is a clearly defined breeding season for parrots in Colombia? As this may influence bird’s counts (females may incubate eggs, or after breeding more young could be observed etc.) it will be useful to provide such information.
- 242 – how variable were prices provided by pet owners, as only average data is presented in the Table? How many such data were obtained from local people?.
Results.
Fig. 3 – add trend line. What means the particular point in the figure – a species? It is mentioned that these are averaged opinions, but no variability is presented. Probably for the sake of clarity, but maybe somewhere variability could be mentioned.
- 226 – four species are mentioned with reference to Table 1, but in Table 1 there is no information for which species the index was not calculated. Also according to Table 2 it seems that only for three species the index could not be calculated.
Tables and figures – It seems that bird species are ordered alphabetically. Authors should consider whether systematic presentation, or maybe in relation to the factors analysed could be a better option.
Fig 6 – the caption is quite long. Probably species name abbreviations could be presented in one of the tables, and in this figure caption may refer to the table.
Discussion
Point 4.1. This point could be shortened, as in many places repeats obtained results
L 355 I’m not sure whether word ‘guerrilas’ is clearly understandable
Appendix. Probably not only number of individuals kept by peoples, but also number of persons keeping them will be useful.
Reviewer 2 Report
Dear authors,
this is a very interesting topic, however it will need some work before publication and I have made extensive comments in the attached pdf. my concerns were mostly about the description of the methodology, the relevance of some sections in the discussion (deleting one and suggesting a different section) and more international context, referring to the Asian songbird trade. I have recommended several articles for your attention. I also suggest remaking the figures and deleting the appendix. some ideas were underexplored and deserve more attention, while others were repeated multiple times and could be simplified or ignored. also, I was asking many times questions that got answered further down. in these cases, you might consider reorganizing the ideas.

Round 2
Reviewer 2 Report
Dear authors, thank you for addressing my comments, I think the article has improved a lot. I am satisfied with the content, however, the MS still needs editing, see suggestions in the attached pdf.

Author Response
Dear authors, thank you for addressing my comments, I think the article has improved a lot. I am satisfied with the content, however, the MS still needs editing, see suggestions in the attached pdf.
R. Thank you very much for the very detailed editing of our revised version, it has greatly contributed to improving the fluency of the text. We have incorporated nearly all the suggestions, except in a very few cases because they slightly changed the meaning of our statements, or in Table 2 because the units (cm and USD) are already indicated in the legend.
As all the minor suggestions deal with the use of proper words or slight rewordings to gain clarity or improve the English, we do not indicate here the numerous lines where we have incorporated the changes. However, all changes can be easily found using the Word track changes function.